# High-Carbohydrate Energy Intake During a Round of Golf-Maintained Blood Glucose Levels, Inhibited Energy Deficiencies, and Prevented Fatigue: A Randomized, Double-Blind, Parallel Group Comparison Study

**DOI:** 10.3390/nu16234120

**Published:** 2024-11-28

**Authors:** Yosuke Nagashima, Kiyohiro Ehara, Yoshitomo Ehara, Ayana Mitsume, Yuhei Uchikoba, Shigeru Mineo

**Affiliations:** 1Department of Health Science, Musashigaoka Junior College, 111-1 Minamiyoshimi, Yoshimi 355-0154, Saitama, Japan; mitsume-a@musashigaoka.ac.jp; 2Department of Education, Tamagawa University, Machida 194-8610, Tokyo, Japan; ehrky0ot@stu.tamagawa.ac.jp; 3College of Sport and Wellness, Rikkyo University, Niiza 352-8558, Saitama, Japan; y-ehara@rikkyo.ac.jp; 4Candy, Dessert & Chilled Product Development Section, Second Product Development Department, Bourbon Corporation, Kashiwazaki 945-8611, Niigata, Japan; uchikoba-yuh@bourbon.co.jp; 5Nutraceuticals Science Laboratory, Advanced Research Institutes, Bourbon Corporation, Kashiwazaki 945-8611, Niigata, Japan; mineo-shi@bourbon.co.jp

**Keywords:** carbohydrates, golf, interstitial glucose, isomaltulose, sport nutrition

## Abstract

Objectives: This study primarily aimed to examine the optimal amount of carbohydrates in the effects of high-isomaltulose and high-sucrose ingestion compared with low-sucrose ingestion on blood glucose levels. The secondary objective was to assess the changes in blood glucose levels that may impact golf-related performance. Methods: This study included 29 healthy male competitive golfers playing 18 holes. These participants were randomly assigned to the low-sucrose (LSUC, 30.9 g/h of carbohydrates), high-sucrose (HSUC, 44.2 g/h of carbohydrates), and high-isomaltulose (HISO, 44.5 g/h of carbohydrates) groups. They were required to continuously consume the test food during the round. Assessment items included blood glucose, golf performance, urinary urea nitrogen, subjective ratings (concentration, fatigue, and relaxation), and anxiety ratings. A main effect of the test meal of changes in interstitial glucose concentration was determined. Results: The HSUC had significantly more interstitial glucose than the HISO and LSUC, while the HISO group had a much lower decrease in urine urea nitrogen than the LSUC group. In subjective fatigue, the main effect of the test diet was observed, and the HSUC and HISO showed significantly lower values than the LSUC after 18 h. Conclusions: Compared with low-carbohydrate intake, high-carbohydrate intake during a round of golf-maintained the blood glucose levels and prevented fatigue. Therefore, this study indicates that competitive golfers need a high-carbohydrate intake of approximately 44 g/h for energy intake during a round of golf.

## 1. Introduction

Golf is classified as a moderate-intensity sport, characterized by a physical activity level of 4.8 metabolic equivalents. A typical round consists of 18 holes, lasting approximately 4 to 5 h, during which participants cover distances exceeding 10 km [1,2,3,4,5,6]. In a recent scoping review, 1936 kcal (range: 531–2467 kcal) was reported for the total energy expended across a round of golf [7]. In competitive play, golf necessitates crucial shot-making decisions, several high-effort swings, putting, and extensive walking during the round, leading to physical weariness, mental exhaustion, and golf-specific fatigue, which adversely impacts performance [8,9]. Carbohydrates can mitigate declines in blood glucose levels, postpone fatigue, and positively influence the central nervous system, thereby enhancing exercise performance [10,11]. The American College of Sports Medicine (ACSM) advises endurance athletes to participate in exercise exceeding 1 h and to ingest carbohydrates at a rate of 30–60 g/h [12]. However, these guidelines may be adapted for golfers during competition. Carbohydrate intake methods have been examined to prevent performance loss during golf competition [1,4,13,14]. In a recent study, we discovered that a continuous carbohydrate ingestion of 30 g/h was effective in reducing fatigue and preserving the glucose concentration of competitive golfers during a competitive 18-hole event [14]. However, an open-study design was employed in the execution of this investigation; therefore, we could not determine whether or not continued carbohydrate intake has any effect on cognitive performance in a blinded study design. In addition, 30 g/h of carbohydrates was consumed as a test meal, but the optimal amount of carbohydrates required for competitive golfers remains unknown. Therefore, the present study sought to address these issues.

To address the first issue, that is, the amount of carbohydrates needed to provide energy for competitive golfers, we conducted a double-blind method using the test meal from the previous study (30.9 g/h) as the reference meal and a test meal with a higher carbohydrate content (44.5 and 44.2 g/h, respectively). Consuming a low-glucose meal before exercise reportedly increases fat oxidation during endurance exercise and improves athletic performance compared with consuming a high-glucose meal [15,16,17,18]. Thus, for the endurance athlete, the type of carbohydrates consumed appears to be a crucial factor. Therefore, we focused on isomaltulose. A structural isomer of sucrose made up of glucose and fructose, isomaltulose is a naturally occurring disaccharide. Isomaltulose has the same energy value as sucrose (4 kcal/g) since it is fully broken down and absorbed in the small intestine. The digestion and absorption rate of isomaltulose is roughly one-fifth that of sucrose [19]. Consequently, isomaltulose consumption results in a diminished elevation in blood glucose and a decrease in insulin production compared to sucrose consumption. Furthermore, the glucose supply is sustained for an extended period of time due to the ability of isomaltulose to prevent abrupt decreases in postprandial blood glucose concentrations that occur after a rapid increase [20,21]. Specifically, golf can generate a longer playing time intake and, most likely, a higher energy expenditure than other sports. Therefore, the consumption of isomaltulose may be particularly advantageous for competitive golfers. Therefore, this study prepared test meals with different carbohydrate contents to examine the optimal amount of sugar in competitive golfers and prepared test meals containing isomaltulose to assess the quality of carbohydrates and its effects.

This study aimed to evaluate the impact of varying carbohydrate contents and compositions on performance in golf. Thus, our primary objective was to investigate the optimal amount of carbohydrates in the effects of high-isomaltulose and high-sucrose ingestion compared with low-sucrose ingestion on blood glucose levels. As the secondary objective, we assessed how the possible blood glucose level changes may impact golf-related performance. We hypothesized that compared with the low-sucrose group, the high-sucrose and high-isomaltulose groups can maintain higher interstitial glucose levels, attenuate the decline in subjective fatigue, and improve golf-related performance. Furthermore, the high-isomaltulose group would stabilize the interstitial glucose level and have a positive impact on subjective fatigue and golf-related performance compared with the high-sucrose group.

## 2. Materials and Methods

### 2.1. Study Design

Figure 1 illustrates the study design. This study utilized a double-blind methodology, with participants randomly assigned to one of the feeding conditions according to a block randomization design. We used age, height, weight, and the mean score of the last five rounds as the stratification factor. Subsequently, using the stratified block randomization method, we randomly assigned the participants to three groups (high-isomaltulose gummies, HISO; high-sucrose gummies, HSUC; and low-sucrose gummies, LSUC). Table 1 shows the physical characteristics of the participants in each group; no significant difference between the assigned groups was confirmed.

The participants’ golf-related performance was assessed in one day at the Ome Golf Club at Ome City, Tokyo. The participants played 18-hole rounds on the test day while we measured their interstitial glucose, golf performance, anxiety, and self-reported cognitive levels and sampled spot urine. This study adhered to the guidelines outlined in the Declaration of Helsinki, and all procedures received approval from the Ethics Committee of Musashigaoka Junior College, Japan (No. 23-2, 11 July 2023). This study was registered in a public database established by the University Medical and Information Network (Study ID: UMIN000052039).

### 2.2. Test Meals

The test meals were three types of gummies prepared by Bourbon Corporation (Niigata, Japan) and packed in nontransparent bags (Figure 1). All of these foods were identical in color and appearance. Carbohydrate intake of test means was 222.3, 221.0, and 154.7 g for HISO, HSUC, and LSUC. The advised carbohydrate consumption during endurance events is 30–60 g/h [12]. If the time required to complete 18 holes in a golf competition is 5 h, 150–300 g of carbohydrates is required. Thus, all test meals in this study were compliant with the guidelines [12].

### 2.3. Study Population

In this study, all 36 male competitive golfers belonging to the second division of the Kanto Student Golf Federation were selected for convenience. The university’s athletic coaches received the study’s recruitment information. In order to obtain a power of 0.80 and an alpha value of 0.05, a minimum of 36 participants (12 per group) were required, as determined by our initial sample size calculation. The inclusion criteria were as follows: (1) consumption of healthy foods and a nonsmoker; and (2) must have competed in national competition in fiscal year 2022; (3) no bone or joint disorders (e.g., back, knee, or hip) that could limit the capacity to play. Conversely, the exclusion criteria were the following: (1) being allergic to the tested food and (2) taking medicines or dietary supplements that can affect the blood glucose levels. The informed consent form was read and signed by the participants, who were also informed about the experimental procedures. Thirty participants consented to participate in this investigation, and ineligibility was at the discretion of the principal investigator.

Thirty-one Japanese competitive golfers who satisfied the inclusion and exclusion criteria provided written informed consent (response rate, 100.0%). However, on the day of the test, one participant declined to participate because of fever. In total, 30 participants (10 in each group) completed all measurements; however, one was excluded from the analyses due to the absence of an interstitial glucose level measurement. Thus, data from 29 participants (10 in HISO, 10 in HSUC, 9 in LSUC) were analyzed (93.5%). Ultimately, these 29 participants (93.5%) were included in the analysis.

### 2.4. Procedures

This research was performed at the Ome Golf Club in Ome, Tokyo, Japan, on 11 September 2023, commencing at approximately 6 AM. The participants were instructed to refrain from alcohol consumption the evening prior to the round and to arrive at the venue without having breakfast on the test day. All participants arrived at the golf course in a fasted state on the testing day and had abstained from alcohol consumption for the preceding 12 h. They were supplied with a device that monitored the interstitial glucose levels at 6:30 AM. Between 7:30 and 8:30 in the morning, it was customary to eat a standardized breakfast of rice, miso soup, salted salmon, nattō, and boiled egg, with 580 calories, 22.2% protein, 17.7% fat, and 63.9% carbohydrates. The Ome Golf Club consistently served a standardized breakfast, and the order of consumption was determined by the time of day. A nutritionist used the weighing method to determine the breakfast’s energy and nutritional content. Japan’s Standard Tables of Food Composition, 2020 edition (8th re-vision), and Eiyou-plus version 1 (Kenpakusha Co., Ltd., Tokyo, Japan) were used to estimate nutritional intake. According to the ACSM guidelines, 1–4 g of carbohydrates per kilogram of body weight (BW) should be consumed 1–4 h prior to exercise [12]. Carbohydrate content was 92.8 g, or 1.38 kg/body weight, in the standard breakfast. This value met the ACSM standard.

After breakfast, the examinees performed their own preparatory exercises, and then each of them practiced hitting and putting 30 balls to prepare for the round of golf. The participants were expected to consume all six of the 500 mL bottles of water they brought. They were only allowed to eat and drink the test food and water that were assigned to each hole until the round was over. Every participant kept track of how much of the test meal they consumed. The CHO intake requested them to chew well, aiming for 30 bites per mouthful. For this study, they played 18 holes on foot, but golf equipment was transported in a cart. To be competitive in this study, reference was made to previous research [22]; we offered gift cards of JPY 3000 to those with the best scores, JPY 2000 to those with the second-best scores, and JPY 1000 to those with the third-best scores.

The first group started at 9:00 AM and the last group started at 10:00 AM. To prevent dehydration, the participants took a 40 min break after completing nine holes. We recorded the start and end times. The first group finished the 18 holes at 2:32 PM, and the last group finished at 3:45 PM. Excluding breaks, the mean round time for this study was 5 h and 2 min. Thus, the mean hourly carbohydrate intake of the participants was 44.5, 44.2, and 30.9 g/h for the HISO, HSUC, and LSUC.

### 2.5. Measures and Data Collection

#### 2.5.1. Interstitial Glucose

The glucose concentration in the blood is closely correlated with the glucose level in the interstitial fluid [23]. We employed interstitial glucose to assess blood glucose levels. The CGM (Freestyle Libre; Abbott Diabetes Care, Alameda, CA, USA) was used to measure the interstitial glucose levels. This technique incessantly quantifies the glucose concentration in the interstitial fluid obtained from cells situated just beneath the skin; its specifics have been documented elsewhere [24,25]. Herein, changes in interstitial glucose, mean interstitial glucose, and coefficient of variation for each test meal during a round of golf were determined and evaluated.

#### 2.5.2. Golf Performance

Golf performance was assessed through five tests: score, driving, iron play, chipping, and putting accuracy. Next, the scores were evaluated to ascertain the discrepancy between the participant’s number of strokes and the predetermined number of strokes for each hole. Strokes were recorded by markers and reported by the participants after checking for errors with the research staff. Driver accuracy was evaluated by the probability of tee shots held in the fairway using a driver. Furthermore, iron accuracy was evaluated by the rate of “par on”, which was defined as hitting the green within two shots of par and was conducted over 18 holes. Recovery was defined in this study as achieving a score of par or greater on a hole that is not par. The recovery rate refers to the frequency with which a player achieves par or better on a hole that is not designated as par. Participants were permitted to use any club for chipping around the green given that multiple clubs could be employed for this purpose. The average number of putts, calculated by counting the number of putts made on holes with par on, was used to evaluate the putting accuracy.

Golf performance was recorded by item and at the points noted. Scores, iron accuracy, and putting performance were evaluated according to each hole. Driver accuracy was on the 14 par 4 and par 5 holes. Chipping accuracy was on the not-par-on holes.

#### 2.5.3. Energy Deficiencies

Energy deficiencies were evaluated according to the urinary urea nitrogen levels. When energy is in short supply, protein is used as energy source, thereby increasing the blood urea nitrogen level [26]. In healthy adults, approximately 55% of blood urea nitrogen is excreted in the urine. Additionally, sex, age, height, weight, and renal function reportedly influence urinary nitrogen excretion [27]. Therefore, we limited the participants to healthy male college students and provided them with a standardized breakfast to assess energy deficiency caused by urinary urea.

In advance, the participants were verbally explained the urine collection method. They were provided with 10 mL tubes for urine collection. Urine samples were obtained on two occasions (once prior to the trial and once following it). All specimens were stored under refrigeration and dispatched to SRL Corporation (SRL, Tokyo, Japan). Urinary urea nitrogen (g/dL) was analyzed using a selective urease-UV method.

#### 2.5.4. Self-Perceived Evaluation

A visual analog scale (VAS) questionnaire that was adopted and utilized in earlier research was employed to conduct self-perceived evaluation [28]. The subsequent three items were assessed: self-perceived levels of concentration (PLC), self-perceived levels of fatigue (PLF), and self-perceived levels of relaxation (PLR). The VAS questionnaire was filled out by the participants. It comprises words that describe the least and highest status indicated at the left and right ends of a horizontal line that is 100 mm long. Every three holes (before the start and after the third, sixth, ninth, twelfth, fifteen, and eighteenth holes), we performed the self-perceived evaluation.

#### 2.5.5. Anxiety

Anxiety was evaluated using the STAI-Y, a 20-item self-reported questionnaire that measures trait and state anxiety. State anxiety reflects the level of anxiety a person is experiencing at the moment, while trait anxiety refers to a person’s predisposition to experience anxiety over a prolonged period. The following 4-point rating system was used to evaluate each question: one represents “almost never”, two “sometimes”, three “often”, and four “almost always” [29]. The Japanese version was employed in this investigation, and it was translated and validated by Hidano et al. [30]. Trait and state anxiety were analyzed using the total score of each item (minimum: 20 points; maximum: 80 points). Furthermore, the survey items were assessed based on their reliability by calculating the Cronbach’s alpha coefficient and utilizing the items that had internal consistency confirmed. The results were 0.775 and 0.760, respectively. Therefore, all items were used in our survey, and 10 items were used as scale items.

#### 2.5.6. Other Variables

The initial collected data were age, height, weight, and the mean score for the last five rounds. This mean the score included noncompetitive golf. All items were addressed through self-report. Ambient temperature, wind speed, wind direction, and precipitation were documented for each trial date. Data were obtained from the Ome Meteorological Agency Observatory, which is situated in close proximity to the golf course [31]. No rainfall was recorded during the trial date.

#### 2.5.7. Statistical Analyses

All data are presented as the mean ± standard deviation. JMP version 14.3.0 (SAS Institute Inc., Cary, NC, USA) was employed to analyze all statistical data. The Shapiro–Wilk test was employed to assess the normality of the data. A two-way analysis of variance (trial × time) with repetition was employed to compare measurements across trials for changes in interstitial glucose concentrations and self-perception assessment. The effect magnitude was quantified using the eta squared (η^2^) as a metric for the two-way analysis (small, 0.01–0.08; medium, 0.08–0.26; large, ≥0.26) [32]. Post hoc analyses utilizing the Tukey test were conducted following the identification of the main effect of enforcement. The evaluation of self-perception, the coefficient of variation in interstitial glucose, and the changes in urinary urea nitrogen were analyzed using the Tukey test. The Tukey test was conducted to determine Cohen’s d, which is a measure of effect size (small <= 0.2; medium, 0.5; large ≥ 0.8) [33]. All *p*-values that were reported were two-tailed, and *p*-values that were lower than 0.05 were deemed statistically significant.

## 3. Results

### 3.1. General Results

The maximum and minimum temperatures on the test day were 33.2 °C and 22.4 °C, respectively, which were slightly higher than the average maximum and minimum temperatures of 31.6 °C and 21.6 °C, respectively, for September. The mean wind speed and direction were 1.7 mph and south, respectively [31]. The recorded putting green conditions for the trial included a cutting height of 4.0 mm, a green speed of 8.5 ft, and a compaction of 23 kg/cm^2^.

### 3.2. Interstitial Glucose

Interstitial glucose concentrations exhibited significant effects for both trial (*p* < 0.001) and time (*p* < 0.001), while the interaction was not significant (*p* = 0.883; η^2^ = 1.604) (Figure 2a). The HSUC continued to be significantly higher than the LSUC from 225 to 330 min and from 390 to 450 min. The range of effect sizes from 225 to 330 min was d = 1.22 to 2.65, with a median of d = 1.41. The range of effect sizes from 390 to 450 min was d = 1.34 to 1.79, with a median of d = 1.69. The HISO was significantly higher than the LSUC at 315 min (*p* = 0.009; d = 1.35) and significantly lower than the HSUC at 255 min (*p* = 0.032; d = 1.12), 270 min (*p* = 0.025; d = 1.23), and 415 min (*p* = 0.023; d = 1.49).

The mean and SD of interstitial glucose of the HISO, HSUC, and LSUC groups was 5.1 ± 0.1, 5.3 ± 0.3, and 5.0 ± 0.4 mmol/L; interstitial glucose of the HSUC was significantly higher than the HISO and LSUC (Figure 2b). The coefficient of variation in interstitial glucose of the HISO was significantly lower than the LSUC (Figure 2c).

### 3.3. Golf Performance

The results indicated significant main effects for the time (trial: *p* = 0.721, time: *p* < 0.001, interaction: *p* = 0.650) (Table 2). Meanwhile, other items showed no significant differences between the groups.

### 3.4. Energy Deficiencies

Urinary urea nitrogen change was significantly lower in the HSUC (*p* = 0.041, d = 1.265) and HISO groups (*p* = 0.009, d = 1.391) than in the LSUC group (Figure 3).

### 3.5. Self-Perceived Evaluation

The differences in PLC and PLR between the groups were not statistically significant (Figure 4a,b). The main effect of the test meal and time was observed in PLF changes (trial: *p* = 0.033, time: *p* < 0.001, interaction: *p* = 0.385; η^2^ = 0.555) (Figure 4c). The HISO and HSUC trials exhibited significantly lower PLF changes than the LSUC trial (respectively, HISO: *p* = 0.017, d = 1.508, HSUC; *p* = 0.015, d = 1.354).

### 3.6. Anxiety

Trait and state anxiety did not exhibit any substantial differences between the groups in this investigation (Table 2).

## 4. Discussion

In this study, high carbohydrate intake maintained the participants’ higher glucose levels and attenuated energy deficiencies compared with low-carbohydrate intake. However, the carbohydrate content was not related to golf performance. Furthermore, in terms of carbohydrate composition, the isomaltulose group showed suppressed blood glucose fluctuations during the round but no significant difference from the sucrose group in golf-related performance. To the best of our knowledge, this study is the first to demonstrate that competitive golfers on an actual golf course using a blinded design can maintain blood glucose levels and attenuate energy deficiencies by consuming high contents of carbohydrates during a round of golf, with respect to low-carbohydrate ingestion.

The primary effect of the test diet on blood glucose was observed (Figure 2a). The test food used by the HISO and HSUC groups contained carbohydrates (44.5 and 44.2 g/h, respectively); this amount is higher than that in LSUC group (30.9 g/h) and our previous cross-over trial (31.1 g/h). The alteration in urine urea nitrogen was greatly reduced in the HSUC and HISO trials compared to the LSUC trial (Figure 2). During exercise, adequate transport of glucose to active muscles is essential to prevent supply deficits. In cases of insufficient energy, protein is utilized as an energy source, resulting in elevated blood urea nitrogen and urinary urea nitrogen levels [26,27]. This study indicates that a high-carbohydrate energy intake can supply the necessary energy for golfing activities. Therefore, consuming more carbohydrates (approximately 44 g/h) helped competitive golfers avoid energy deficits and stabilize blood glucose levels during the round. Hence, this study’s results show that competitive golfers should consume more carbohydrates (approximately 44 g/h) during golf play. Moreover, this study found that high carbohydrate intake reduced the increase in perceived fatigue on the 18th hole compared with low carbohydrate intake (Figure 4c). Several studies have focused on the relationship between energy intake during golf and fatigue [4,13,14]. In our previous study, continuous carbohydrate intake (31.1 g/h) reduced fatigue and maintained concentration compared with no intake [14]. Stevenson et al. conducted a study involving 20 male golfers (mean ± standard deviation: age 23 ± 4 years, stature 176.4 ± 5.6 cm, mass 72.8 ± 17.4 kg, and handicap 15 ± 4). The administration of 0.64 g/kg body mass of carbohydrates prior to and twice during the round at holes 6 and 12 enhanced both motor performance (assessed through 2 and 5 m putts) and cognitive performance (self-reported alertness and fatigue scores) [4]. Thompsett et al. reported another study where the participants were males aged between 14 and 65 years, with a USGA handicap of ≤18, and with a carbohydrate energy intake of 30 g, carbohydrate + protein energy intake of 15 g each, and a 0 kcal control before the start of the game. It was reported that the intake of carbohydrates and carbohydrates + protein in the second half of the nine holes was beneficial for improving golf performance and maintaining fatigue levels [22]. In our previous study, 11 male golfers participated, with a mean age of 20 ± 1 years, stature of 173.6 ± 5.9 cm, mass of 74.8 ± 13.7 kg, and body fat percentage of 21.6 ± 6.4%. Continuous carbohydrate intake was measured at 31.1 g/h. Fatigue was reduced in comparison to no intake after the 6th to 18th holes [14]. No studies have specifically examined the effects of high or low carbohydrate intake on competitive golfers; our results show that high carbohydrate energy intake may suppress fatigue compared with low carbohydrate intake. Our results thereby indicate a new finding in the dietary strategy of competitive golfers.

Analysis of carbohydrate composition revealed that the coefficient of variation for interstitial glucose in the isomaltulose group was significantly lower than that observed in the low-sucrose group, indicating that the isomaltulose group exhibited the least variation (Figure 2c). A previous study reported that isomaltulose is less likely to elevate blood glucose levels, is insulin-hypoallergenic, inhibits the elevation in blood glucose level from other carbohydrates, and avoids hypoglycemia through blood glucose stabilization [20,21,34]. The effect of isomaltulose ingestion on blood glucose fluctuations before and after exercise during high-intensity exercise has been studied [35]. However, to the best of our knowledge, no studies have examined the impacts of moderate-intensity activities like golf. Our results showed that isomaltulose ingestion stabilizes blood glucose levels compared with sucrose ingestion during a medium-intensity, prolonged golf competition. This study also presents a new finding in nutritional strategy during golf play. Interestingly, when looking at the readings from 280 to 330 min (11:30 to 12:30), the LSUC group experienced a substantial decrease in blood glucose levels, while the HISO group experienced a gradual decrease. In general, high blood glucose levels and sudden fluctuations in these levels are associated with decreased concentration and drowsiness [36]. In addition, competitive golfers often complain of drowsiness after the middle of a round. Further studies are warranted to clarify the association between high blood glucose levels and sudden fluctuations in blood glucose levels and drowsiness during golf play.

In this study, differences in the quantity and quality of carbohydrates did not significantly affect golf performance (Table 1). Hypoglycemia may result in diminished concentration, increased irritability, suboptimal decision-making, and elevated assessments of self-reported fatigue [11]. Prior research indicates that the consumption of carbohydrates while playing golf may enhance performance on the course [4,22]. On the Professional Golfers’ Association of America Tour, the scores can be accounted for by driving distance, fairway percentage, par on percentage, and shots made, with a breakdown of 61–87% [37,38]. Based on these data, we expected that mitigating fatigue would avert a deterioration in driving performance. Nonetheless, the findings of this study are inconsistent with those of earlier research. There are several potential explanations. Initially, it is conceivable that this study failed to sufficiently induce weariness in the individuals. The college golfers—participants of this test—moved their caddy bags during practice, but in this test, they were carried in a cart; therefore, they may not have been sufficiently fatigued because they were not moving around as much. Second, the time was possibly too short for fatigue to have an effect on golf performance. Golf is a sport characterized by competition, typically spanning a duration of four days. Therefore, maintaining a medium- to long-term condition is essential. Lack of energy during competition can affect conditioning the following day and beyond. In recent years, the relationship between energy deficit during exercise and nocturnal hypoglycemia has been investigated [39]. Nocturnal hypoglycemia might lead to sleepiness during exercise, as well as decreased concentration and skill, because it releases noradrenalin, which promotes arousal and leads to poor sleep quality. From this perspective, in the future, the effects of differences in the quantity and quality of carbohydrates during golf play on fatigue and golf performance on the following day and beyond should be examined.

This study’s findings could enhance the nutritional strategies employed by competitive golfers. Competitive golfers may play 18 competitive golf holes while carrying their own luggage, necessitating a greater energy expenditure during the competition. Kasper et al. recently reported that golfers consume more energy when carrying their own bags than when using manual and electronic trolleys [7]. A total of 64 Japanese elite junior golfers (32 male participants and 32 female participants) ingested carbohydrates at a rate of 26.1 ± 13.8 g/h for 18 holes, as reported in our previous research [40]. This is less than the low carbohydrate energy intake in this study, and more carbohydrates need to be consumed to maintain blood sugar levels and suppress fatigue. Therefore, competitive golfers should plan a nutritional strategy to increase carbohydrate intake to prevent hypoglycemia and control fatigue during a round of golf.

Regarding this study’s strengths, this study used a double-blind study design. It was also conducted on an actual golf course with competitive golfers while continuously measuring the interstitial glucose levels and biological outcomes. However, this study also has several limitations.

First, it has a small number of participants. A sample of at least 36 participants (12 per group) was necessary to achieve a power of 0.80 and an alpha value of 0.05; however, in this study, we were only able to recruit 30 participants. Regardless of this study’s outcome, small sample numbers are troublesome; problems can range from unrepeatable “discoveries” to inconclusive results and low precision [41]. Second, the test food (30.9 g/h) that was effective in the previous test was used as the standard for the present study. Moreover, the lack of a true placebo was a limitation of this study. Placebo gummies with noncaloric artificial sweeteners were not considered ethical to be used in this study because they exceeded the maximum no-action dose. Third, golf performance was set as the main outcome, but there are various confounding factors (personal characteristics, emotions, golf skills, and physical condition) that affect golf performance. In this study, we took into account the differences in golf skills between the groups, but we were unable to take into account factors such as emotions and physical condition. Fourth, the CGM systems are subject to an inescapable time delay, which is pertinent during the exercise. The employed CGM device exhibits a slower rise time compared to blood sampling and typically yields lower peak glucose levels, perhaps underestimating the impact of carbohydrate consumption on the glucose response [42]. Fifth, urinary urea nitrogen has many confounding factors, including sex, age, height, weight, and diet [26,27]. We considered these confounding factors while conducting this study; however, we cannot confirm whether we could eliminate all of them. Hence, the reliability of the evaluation of energy deficiency using urinary urea nitrogen was not considered high. Sixth, although significant effects were also found in subjective fatigue, this study did not assess it using in vivo fatigue markers.

## 5. Conclusions

In contrast to low carbohydrate intake, high carbohydrate intake during a round of golf maintained the golfers’ blood glucose levels, suppressed energy deficits, and prevented fatigue. Therefore, high carbohydrate intake may be effective in maintaining the blood glucose levels and attenuating fatigue in competitive golfers, and these results help suggest dietary plans for competitive golf.

Although isomaltulose showed stable blood glucose fluctuations, no significant difference was noted between isomaltulose and sucrose.

## Figures and Tables

**Figure 1 nutrients-16-04120-f001:**
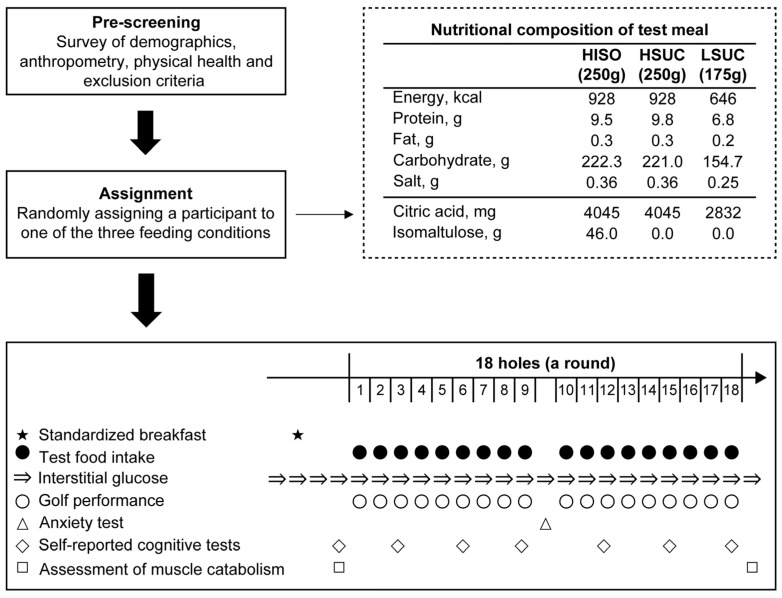
Study design.

**Figure 2 nutrients-16-04120-f002:**
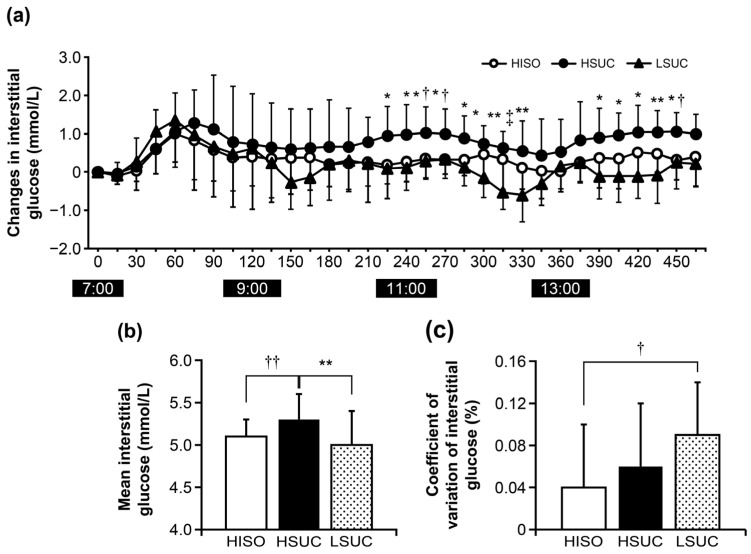
(**a**) Impact of a test meal on how interstitial glucose levels change throughout a game of golf, (**b**) mean interstitial glucose, (**c**) coefficient of variation in interstitial glucose. Data are expressed as means ± standard deviation. Measurements across trials were analyzed using a two-way repeated measure analysis of variance. Measurements of interstitial glucose concentration variations between trials were analyzed using a two-way analysis of variance (trial × time) with repetition. When significant differences were found, differences between trials were identified using the Tukey test. Mean interstitial glucose and self-perceived evaluation were conducted using the Tukey test. Significant difference between HSUC and LSUC trials (* *p* < 0.05, ** *p* < 0.01). Significant difference between HSUC and HISO trials (^†^ *p* < 0.05, ^††^ *p* < 0.01). There is a significant difference between the HISO and LSUC trials (^‡^ *p* < 0.05). HISO denotes the high-isomaltulose group; HSUC refers to the high-sucrose group; LSUC indicates the low-sucrose group.

**Figure 3 nutrients-16-04120-f003:**
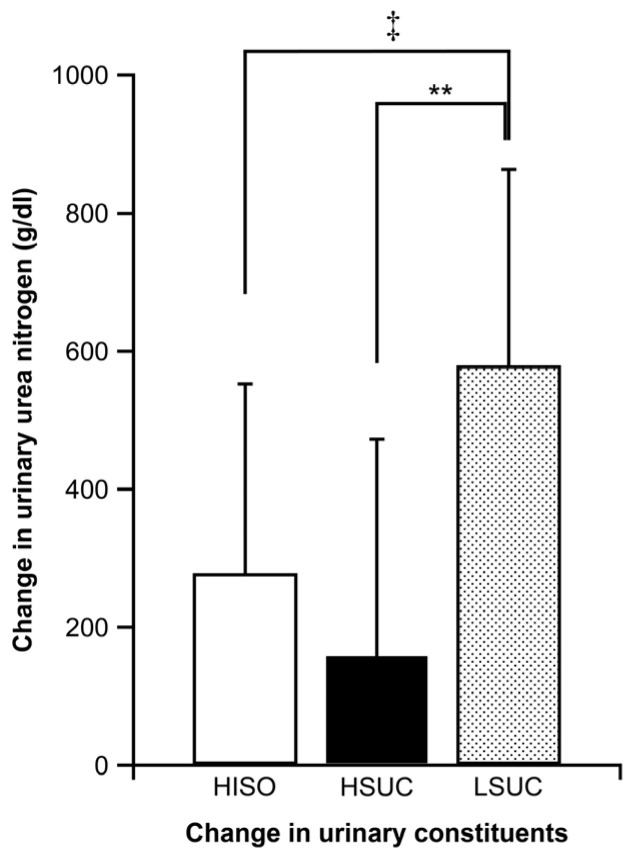
Changes in urinary urea nitrogen. Data are presented as means ± standard deviation. The Tukey test was employed to analyze measurements across trials. There is a notable distinction between the HSUC and LSUC trials (** *p* < 0.01). There is a notable distinction between the HISO and LSUC trials (^‡^
*p* < 0.01).

**Figure 4 nutrients-16-04120-f004:**
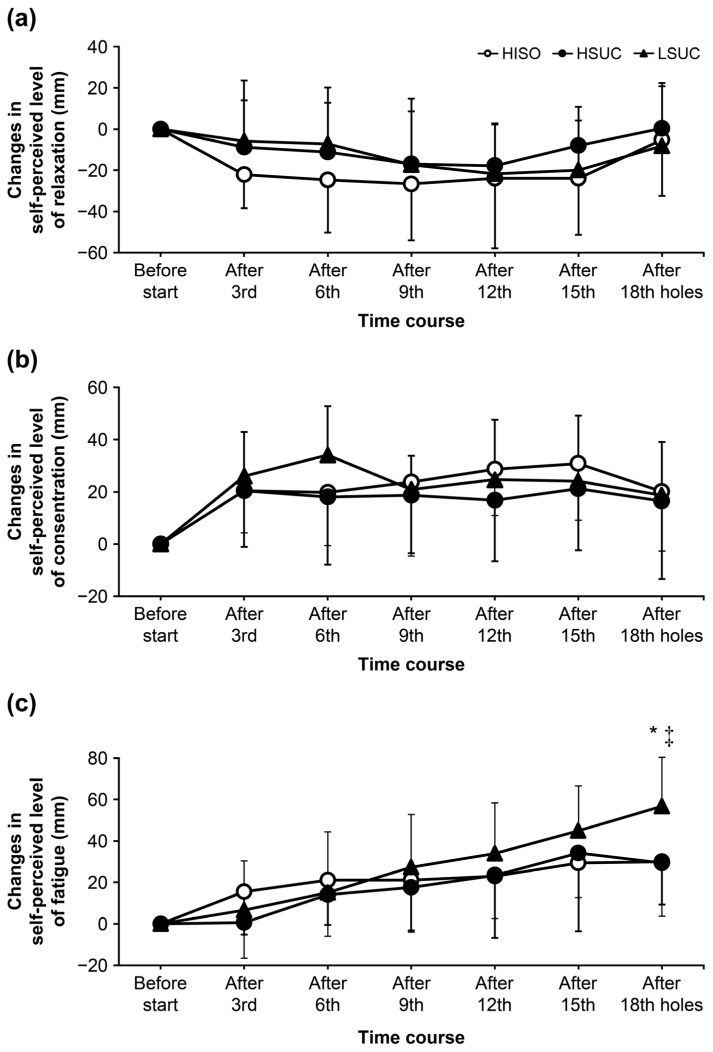
Changes in self-perception. (**a**) Changes in the self-perceived level of relaxation, (**b**) alterations in self-assessed concentration levels, (**c**) alterations in the self-assessed degree of fatigue. Data are presented as means ± standard deviations. A two-way repeated measure analysis of variance was employed to compare measurements across trials. The interaction effect of trial and time was evaluated. When substantial differences were identified, the Tukey test was employed to discern variations between trials. There is a significant difference between HSUC and LSUC trials (* *p* < 0.05). There is a significant difference between the HISO and LSUC trials (^‡^ *p* < 0.01).

**Table 1 nutrients-16-04120-t001:** Physical characteristics of the study participants.

	HISO (*n* = 10)		HSUC (*n* = 10)		LSUC (*n* = 9)	
	Mean	SD	Mean	SD	Mean	SD
Age, years	19.9	1.0	20.3	1.3	20.8	1.0
Height, cm	172.3	4.1	172.9	5.3	172.6	5.2
Weight, kg	68.7	6.6	66.7	6.3	66.0	8.7
BMI, kg/m^2^	23.1	2.1	22.4	2.2	22.1	2.2
Mean score in the last five rounds, strokes	77.4	8.6	79.0	9.2	78.3	8.3

SD, standard deviation; BMI, body mass index.

**Table 2 nutrients-16-04120-t002:** Golf performance measurements and STAI score for HISO, HSUC, and LSUC trials.

	Trials	Time Course (Holes)		Main Effect		Interaction
		1–9 Holes	10–18 Holes	Trial	Time	
				*p*	*p*	*p*
Golf performance						
Scores (strokes)	HISO	40.4 ± 6.2	32.9 ± 2.3	0.721	<0.001	0.650
	HSUC	39.3 ± 4.9	34.4 ± 2.1			
	LSUC	38.6 ± 4.7	32.9 ± 5.5			
Mean putting scores (strokes)	HISO	1.68 ± 0.31	1.65 ± 0.31	0.252	0.492	0.978
	HSUC	1.87 ± 0.25	1.80 ± 0.32			
	LSUC	1.75 ± 0.20	1.68 ± 0.42			
Driving accuracy (%)	HISO	42.9 ± 25.2	47.2 ± 15.1	0.701	0.588	0.576
	HSUC	42.9 ± 27.2	40.0 ± 24.0			
	LSUC	44.5 ± 20.7	33.3 ± 18.9			
Iron accuracy (%)	HISO	48.9 ± 28.8	51.1 ± 17.6	0.301	0.387	0.878
	HSUC	42.2 ± 21.5	51.1 ± 14.1			
	LSUC	55.6 ± 22.9	59.3 ± 21.5			
Chipping accuracy (%)	HISO	59.8 ± 41.3	69.2 ± 39.3	0.994	0.159	0.779
	HSUC	53.3 ± 34.4	74.1 ± 34.4			
	LSUC	60.0 ± 22.9	66.7 ± 21.5			
STAI score						
State anxiety	HISO	44.0 ± 10.0	-	0.978	-	-
	HSUC	44.7 ± 9.7	-			
	LSUC	44.8 ± 7.5	-			
Trait anxiety	HISO	45.2 ± 8.9	-	0.710	-	-
	HSUC	46.4 ± 8.7	-			
	LSUC	48.2 ± 5.5	-			

The data are displayed as mean ± SD. A two-way analysis of variance with repetition was employed to compare measurements of golf performance across trials. Trail effect and tested for interaction (Trail × Time). Trail effect and tested for interaction (Trail × Time). When significant differences were found, Tukey tests were used to identify differences between trails. Tukey tests were employed to identify differences between trails for the measurements of the STAI Score.

## Data Availability

The datasets generated and/or analyzed during this study are not publicly available because our ethical approval did not include the use of these data by other researchers. The materials pertinent to this study is available from the corresponding author on reasonable request.

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
