# Peer review of "High-Carbohydrate Energy Intake During a Round of Golf-Maintained Blood Glucose Levels, Inhibited Energy Deficiencies, and Prevented Fatigue: A Randomized, Double-Blind, Parallel Group Comparison Study"

_nutrients, 2024, doi:10.3390/nu16234120_

Round 1

Reviewer 1 Report

Comments and Suggestions for Authors

This great article examines carbohydrate needs in professional sportspeople, i.c. golf. Although the sample size is rather small, both the set-up and work-up of this pilot study are very well done, guided by sound statistics. I have some remarks:

Section 3.1: I would propose to put the study subjects' characteristics in a table.

Table 1: maybe some more information is needed (e.g. caloric needs), as the sports may not be widely known to all.

Figure 3: please note that urinary urea nitrogen is fairly unreliable, due to many confounders (hydration, protein intake,...)

Author Response

RESPONSE TO REVIEWER 1

GENERAL COMMENTS: This great article examines carbohydrate needs in professional sportspeople, i.c. golf. Although the sample size is rather small, both the set-up and work-up of this pilot study are very well done, guided by sound statistics. I have some remarks:

RESPONSE: We sincerely thank you for your efforts in providing us with such important insights, which have been helpful in remarkably improving our paper. In the following sections, you will find our responses to each of your points and suggestions. We are grateful for the time and energy you expended for improving our manuscript.

COMMENT 1: Section 3.1: I would propose to put the study subjects' characteristics in a table.

RESPONSE: Accordingly, we have added Table 1 (Lines 97–99 and 108–109).

COMMENT 2: Table 1: maybe some more information is needed (e.g. caloric needs), as the sports may not be widely known to all.

RESPONSE: Accordingly, we have changed the text as follows: "In a recent scoping review, 1,936 kcal (range: 531–2,467 kcal) was reported for the total energy expended across a round of golf [7].” (Lines 38-40)

COMMENT 3: Figure 3: please note that urinary urea nitrogen is fairly unreliable, due to many confounders (hydration, protein intake,...)

RESPONSE: We have added the following text to the Discussion section (lines 480–484): “Fourth, urinary urea nitrogen has many confounding factors, including sex, age, height, weight, and diet [27, 28]. We considered these confounding factors while conducting this study; however, we cannot confirm whether we could eliminate all of them. Hence, the reliability of the evaluation of energy deficiency using urinary urea nitrogen was not considered high.”

OTHERS

Other points to note and additional corrections are described below.

OTHER RESPONSE 1: The similarity index of articles, excluding the references, was 30%. As requested, we had the manuscript rewritten by an experienced scientific editor, who reduced the similarities and improved the grammar and stylistic expression of the paper.

OTHER RESPONSE 2: A minor error in the numbers in the test meals was noted. Therefore, we have revised the panels in Figure 1 and figures in the main text. Given that it was a minor correction, it did not affect the test results or discussion.

Thank you once again for your valuable comments and suggestions. We are hopeful that our supplementary analyses and revised focus have improved your opinion of our work.

Reviewer 2 Report

Comments and Suggestions for Authors

I'am sorry but I'am not able to judge a articel like this. First they published last year a similiar article for now they just changed the "diets".

Overall there is no result on how this changes golf performance, therefore no results no artikel.

Comments on the Quality of English Language

I'am sorry but I'am not able to judge a articel like this. First they published last year a similiar article for now they just changed the "diets".

Overall there is no result on how this changes golf performance, therefore no results no artikel.

Author Response

RESPONSE TO REVIEWER 2

GENERAL COMMENT: I'am sorry but I'am not able to judge a articel like this. First they published last year a similiar article for now they just changed the "diets".Overall there is no result on how this changes golf performance, therefore no results no artikel.

RESPONSE 1: We sincerely thank you for your valuable comments. We have sincerely considered the raised issues. However, the scientific value of this research remains valid. We have conducted a new study using a new study design, which included a placebo, to resolve the issues of the previous study. This test aimed to examine the amount of carbohydrates.

For many years, the need for energy and carbohydrate intake in golf was not studied; however, recently, high energy consumption has been reported. Therefore, our study results, which showed that golfers, similar to other athletes, need to consume energy and carbohydrates, are novel, given that the scope of research has been expanded to include golfers.

OTHERS

Other points to note and additional corrections are described below.

OTHER RESPONSE 1: Regarding the similarity index of articles (excluding the references) being 30%, as requested, we had the manuscript rewritten by an experienced scientific editor, who reduced the similarities and improved the grammar and stylistic expression of the paper.

OTHER RESPONSE 2: A minor error in the numbers in the test meals was identified. Therefore, we have revised the panels in Figure 1 and figures in the main text. Given that it was a minor correction, it did not affect the test results or discussion.

Thank you once again for your valuable comments and suggestions.

Reviewer 3 Report

Comments and Suggestions for Authors

The similarity index of the article, excluding references is 30%. From them, 20% of similarity becomes from the article entitled “Effects of Continuous Carbohydrate Intake with Gummies during the Golf Round on Interstitial Glucose, Golf Performance, and Cognitive Performance of Competitive Golfers: A Randomized Repeated-Measures Crossover Design”, published by the same authors and related with the same trial.

In fact, the manuscript presented here can be considered an extension of the already published article, which from an ethical point of view is at least debatable first publish a small part of the work, and then an expanded version of it.

Overall, the article is well written and logically structured, although it includes some data that in the context of a manuscript on nutrition are somewhat unnecessary. Some of the results, such as golf performance, are very doubtful that they can be attributed to carbohydrate intake. Sports performance is linked to a large number of factors, including individual idiosyncratic, emotional, game-dependent random factors, and so on. Even professional golfers of the highest international reputation have better and worse days with respect to their sporting performance, so I don't think they can be so easily related to carbohydrate metabolism.

The conclusions that the authors mention as results of the work are as follows  “In this study, high-carbohydrate intake during a round of golf-maintained golfers’  blood glucose levels, inhibited energy deficiencies, and prevented fatigue compared with low-carbohydrate intake. Therefore, high-carbohydrate intake may be effective in maintaining the blood glucose levels and attenuating fatigue in competitive golfers, these findings can propose nutritional strategies for competitive golf.” I think they are excessively generic, in fact, they could have been written before the start of the trial, and not only applicable to golf but to any sport.

However, I believe that the biggest problem is that the authors themselves contradict themselves in the wording of the paper, making it clear that the number of subjects used in the study is not sufficient so that, even if the results could be unequivocally related to the metabolism of the carbohydrates administered, the results would be relevant. In page 3, the authors claims: “On the basis of our initial sample 119 size calculation, a sample of at least 36 participants (12 per group) was necessary to 120 achieve a power of 0.80 and an alpha value of 0.05.”. Finally, the number od participants were 29, 9-10 per group. So, the number of participants is below the minimum number that the authors' own statistical analysis considered to be the minimum to achieve statistical relevance

Author Response

RESPONSE TO REVIEWER 3

GENERAL COMMENTS: The similarity index of the article, excluding references is 30%. From them, 20% of similarity becomes from the article entitled “Effects of Continuous Carbohydrate Intake with Gummies during the Golf Round on Interstitial Glucose, Golf Performance, and Cognitive Performance of Competitive Golfers: A Randomized Repeated-Measures Crossover Design”, published by the same authors and related with the same trial. In fact, the manuscript presented here can be considered an extension of the already published article, which from an ethical point of view is at least debatable first publish a small part of the work, and then an expanded version of it.

RESPONSE: We apologize for this oversight. To clarify, we had the manuscript rewritten by an experienced scientific editor who reduced the similarities and improved the grammar and stylistic expression of the paper. We believe that this new information adequately addresses your comment. Details have been omitted due to the large number of corrections.

In addition, we have developed and applied a research design (open trial design) based on the pilot study results. Furthermore, the previous main trial had a different objective, that is, to examine the quantity and quality of carbohydrates. Therefore, it is not the same trial. We apologized for the confusion.

COMMENT 1: Overall, the article is well written and logically structured, although it includes some data that in the context of a manuscript on nutrition are somewhat unnecessary. Some of the results, such as golf performance, are very doubtful that they can be attributed to carbohydrate intake. Sports performance is linked to a large number of factors, including individual idiosyncratic, emotional, game-dependent random factors, and so on. Even professional golfers of the highest international reputation have better and worse days with respect to their sporting performance, so I don't think they can be so easily related to carbohydrate metabolism.

RESPONSE: Accordingly, we have added the following text to the revised manuscript: “Second, golf performance was set as the main outcome; however, it can be affected by various confounding factors, such as personal characteristics, emotions, golf skills, and physical condition. Herein, we considered the differences in golf skills between the groups but not factors such as emotions and physical condition.” (Lines 477–481)

COMMENT 2: The conclusions that the authors mention as results of the work are as follows “In this study, high-carbohydrate intake during a round of golf-maintained golfers’  blood glucose levels, inhibited energy deficiencies, and prevented fatigue compared with low-carbohydrate intake. Therefore, high-carbohydrate intake may be effective in maintaining the blood glucose levels and attenuating fatigue in competitive golfers, these findings can propose nutritional strategies for competitive golf.” I think they are excessively generic, in fact, they could have been written before the start of the trial, and not only applicable to golf but to any sport.

RESPONSE: Thank you for your pertinent comment. Your notes are applicable to golf and other sports, and they may be right. However, the need for energy and carbohydrates in golf has not been widely discussed. The study results, which showed that golfers, similar to other athletes, need to consume energy and carbohydrates, are considered novel because the scope of research has been expanded to include golfers.

However, I believe that the biggest problem is that the authors themselves contradict themselves in the wording of the paper, making it clear that the number of subjects used in the study is not sufficient so that, even if the results could be unequivocally related to the metabolism of the carbohydrates administered, the results would be relevant. In page 3, the authors claims: “On the basis of our initial sample 119 size calculation, a sample of at least 36 participants (12 per group) was necessary to 120 achieve a power of 0.80 and an alpha value of 0.05.”. Finally, the number od participants were 29, 9-10 per group. So, the number of participants is below the minimum number that the authors' own statistical analysis considered to be the minimum to achieve statistical relevance

RESPONSE: We have changed the text to the following: “First, it has a small sample size. A sample of at least 36 participants (12 per group) was required to achieve a power of 0.80 and an alpha value of 0.05. However, we could only recruit 30 participants. Regardless of the outcome, having a small sample size can be an issue; problems can range from unrepeatable “discoveries” to inconclusive results and low precision [43].” (Lines 473–477)

OTHERS

Other points to note and additional corrections are described below.

OTHER RESPONSE 1: A minor error in the numbers in the test meals was noted. Therefore, we have revised the panels in Figure 1 and figures in the main text. Given that it was a minor correction, it did not affect the test results or discussion.

Thank you once again for your valuable comments and suggestions. We are hopeful that our supplementary analyses and revised focus have improved your opinion of our work.

Reviewer 4 Report

Comments and Suggestions for Authors

The study design and use of power calculations is appropriate, the comclusions are in line with the results achieved. The limitations due to number, etc are discussed in the conclusion.

This is a double blind study, I think one improvement that is required is detials around how the subjects were allocated. - coin toss, etc. Other than that it is a useful study and has implications for other sports that have simliar energy expenditures over long periods of time. 

Author Response

RESPONSE TO REVIEWER 4

GENERAL COMMENTS: The study design and use of power calculations is appropriate, the comclusions are in line with the results achieved. The limitations due to number, etc are discussed in the conclusion.

RESPONSE: We sincerely thank you for your efforts in providing us with important insights. We are glad that you think our work will spark debate in our field. Our responses to each of your points and suggestions are shown in the following sections. We are grateful for the time and energy you have expended for the improvement of our manuscript.

COMMENTS 1: This is a double blind study, I think one improvement that is required is detials around how the subjects were allocated. - coin toss, etc. Other than that it is a useful study and has implications for other sports that have simliar energy expenditures over long periods of time.

RESPONSE: We agree with you and have incorporated this suggestion throughout our paper. We have revised the Discussion section as follows (lines 92–97): “This study utilized a double-blind methodology, with participants randomly assigned to one of the feeding conditions according to a block randomization design. We used age, height, weight, and the mean score of the last five rounds as the stratification factor. Subsequently, using the stratified block randomization method, we randomly assigned the participants to three groups (high-isomaltulose gummies, HISO; high-sucrose gummies, HSUC; and low-sucrose gummies, LSUC).”

OTHERS

Other points to note and additional corrections are described below.

OTHER RESPONSE 1: Regarding the similarity index of articles (excluding references) being 30%, as requested, we had the manuscript rewritten by an experienced scientific editor, who reduced the similarities and improved the grammar and stylistic expression of the paper.

OTHER RESPONSE 2: A minor error in the numbers in the test meals was noted. Therefore, we have revised the panels in Figure 1 and figures in the main text. Given that it was a minor correction, it did not affect the test results or discussion.

Again, we wish to thank you for your valuable comments.

Round 2

Reviewer 2 Report

Comments and Suggestions for Authors

test

Author Response

RESPONSE TO REVIEWER 2

RESPONSE: We wish to express our strong appreciation to the reviewers for their insightful comments regarding our manuscript. The comments and suggestions have helped us substantially improve our manuscript.

Reviewer 3 Report

Comments and Suggestions for Authors

Although I believe that the number of subjects is small and the conclusions are somewhat generic, the authors have incorporated explanations in both cases that help the reader to understand the reason for them, so I understand that the manuscript is acceptable for publication in its current state.

Author Response

RESPONSE TO REVIEWER 3

GENERAL COMMENTS: Although I believe that the number of subjects is small and the conclusions are somewhat generic, the authors have incorporated explanations in both cases that help the reader to understand the reason for them, so I understand that the manuscript is acceptable for publication in its current state.

RESPONSE: We wish to express our appreciation to the reviewers for their insightful review of manuscript. The comments have helped us considerably improve the paper.
